# The Cost-Effectiveness of Beta-Lactam Desensitization in the Management of Penicillin-Allergic Patients

**DOI:** 10.3390/antibiotics14070646

**Published:** 2025-06-25

**Authors:** Alicia Rodríguez-Alarcón, Santiago Grau, Silvia Gómez-Zorrilla, Carlos Rubio-Terrés

**Affiliations:** 1Pharmacy Department, Hospital del Mar, 08003 Barcelona, Spain; arodriguezalarcon@hmar.cat; 2Medical School, Universitat Autònoma de Barcelona (UAB), 08193 Barcelona, Spain; sgomezzorrilla@hmar.cat; 3Department of Medicine, Universitat Pompeu Fabra (UPF), 08003 Barcelona, Spain; 4Center for Biomedical Research in Infectious Diseases Network (CIBERINFEC), Instituto de Salud Carlos III, 28029 Madrid, Spain; 5Infectious Diseases Department, Hospital del Mar, Infectious Pathology and Antimicrobial Research Group (IPAR), Hospital del Mar Research Institute, 08003 Barcelona, Spain; 6Health Value, SL., 28034 Madrid, Spain; crubioterres@healthvalue.org

**Keywords:** desensitization, cost-effectiveness, penicillin allergy, antimicrobial stewardship programs

## Abstract

**Background/Objectives:** Antibiotic management of hospitalized penicillin-allergic patients (PAPs) is associated with prolonged hospital stays, adverse reactions and treatment failure, resulting in increased healthcare costs. This study aimed to estimate the cost-effectiveness of beta-lactam desensitization (DES) in the management of PAPs. **Methods**: A cost-effectiveness analysis was performed using a probabilistic model with 1000 s-order Monte Carlo simulations. Hospital costs (in 2025 Euros) and effectiveness outcomes (cure and survival rates) were derived from a Spanish retrospective case–control study conducted between 2015 and 2022, which included 56 PAPs (14 in the desensitization group [DES] and 42 in the control group without DES [NDES]; ratio 1:3), and collected healthcare costs per patient. **Results**: The incremental cost of the DES group was EUR 37,805 (95% CI: EUR 2023–EUR 126,785), with a 100% probability of incurring additional costs compared to the NDES group. The cure rate was 16.5% higher in the DES group (95% CI: 13.3–20.0%), and the estimated gain in life-years per patient (LYG) was 1.42 (95% CI: 1.15–1.73) versus NDES. The cost per life-year gained (LYG) with DES versus NDES was EUR 24,618 ± EUR 19,535 (95% CI: EUR 1755–EUR 73,488). The probability that DES would be cost-effective (cost per LYG < EUR 25,000 and <EUR 30,000) was 61.1% and 100%, respectively. **Conclusions**: According to this analysis, DES appears to be a cost-effective option for managing PAPs. These findings should be confirmed in clinical studies with larger sample sizes.

## 1. Introduction

Penicillin-allergic patients (PAPs) are common in hospital settings, as penicillin allergies labels have an estimated prevalence of 15–20% in the general population [1,2]. However, a significant proportion of these labels are inaccurate or outdated, as true IgE-mediated hypersensitivity is confirmed in less than 10% of cases upon formal allergy testing [3,4]. Despite this, the presence of a penicillin allergy label often leads clinicians to avoid beta-lactam antibiotics, resulting in the use of alternative agents that are broader-spectrum, less effective, or more toxic [1,5]. Prolonged hospital stays and higher mortality rates are present in PAPs due to the failure of alternative treatment [6], antimicrobial resistance selection, or adverse reactions, such as Clostridioides difficile infection, surgical site infections, or adverse events, especially when exposed to nephrotoxic agents like vancomycin or aminoglycosides [1,3,7,8,9,10,11].

The economic burden of hospitalized PAPs is supposed to be higher as some studies have suggested [12,13]. This is attributed to extended lengths of stay, costlier diagnoses, and procedures and the administration of more expensive or toxic antimicrobials. In fact, hospital charges for PAPs have been reported to be approximately 9.1% higher than for non-labeled patients [12].

Desensitization (DES) may be a useful strategy for allergic patients when alternative treatments are suboptimal or unavailable [14,15]. DES is a protocolized process that induces temporary tolerance to the antibiotic through the administration of incrementally increasing doses of a diluted formulation [1,3]. Accordingly, it is possible to safely give beta-lactams to PAPs as suboptimal doses of drugs do not produce immunologic responses to allergens and avoid anaphylaxis due to inhibition of IgE cross-linking and mast cell degranulation [16]. This strategy can effectively and safely allow the use of first-line antibiotics in IgE-mediated allergic patients which poses an interesting option for antimicrobial stewardship programs (ASPs) [14]. In ASPs, one of the main goals is the early de-labeling of PAPs by performing skin tests and oral provocation as these are considered the gold-standard for allergy diagnosis [17,18], but DES is considered a valid option for high-risk patients (history of immediate severe hypersensitivity reaction or in a compromised clinical situation) or those with a confirmed allergy when alternative treatment is not available or is suboptimal and could compromise patient’s life expectancy due to the source of infection [1,17]. However, the implementation of DES protocols as part of ASPs is less widespread [19,20,21].

In 2023, a study was published comparing clinical outcomes between PAPs treated with antibiotic DES and those who received alternative non-beta-lactam antibiotic therapies [15]. This Spanish retrospective case–control study, conducted between 2015 and 2022, included 56 PAPs (14 in the DES group and 42 in the non-desensitized control group [NDES], in a 1:3 ratio). The thirty-day mortality was 14.3% in the DES group versus 28.6% in the NDES group (*p* = 0.24). A clinical cure was achieved in 71.4% of the DES group and 54.8% of the NDES group (*p* = 0.22) [15]. Although these differences were not statistically significant, they suggest a clinically relevant trend favoring DES. These findings point toward potential improvements in patient outcomes, reduced complications, and the possibility of greater cost-effectiveness. Given the mounting pressure on healthcare systems to optimize resource allocation and promote high-value care, assessing the economic implications of DES is particularly timely. To assess this hypothesis, an economic model was developed to estimate the cost-effectiveness of beta-lactam DES in the treatment of PAPs.

## 2. Results

### 2.1. Deterministic Cost and Effectiveness Analysis

The cost per patient with DES and NDES was EUR 53,665 and EUR 14,102, respectively. Therefore, the cost per patient was EUR 39,563 higher with DES (Table 1). The cure rate with DES and NDES was 71.4% and 54.8%, respectively [15]. The survival rate with DES and NDES was 85.7% and 71.4%, respectively (Table 1).

### 2.2. Probabilistic Cost Analysis

The mean incremental cost per patient associated with DES, compared to NDES, was EUR 37,805 (95% CI: EUR 2023; EUR 126,785) (Table 2). The probability that the DES option entails an additional cost was estimated at 100%.

### 2.3. Probabilistic Effectiveness Analysis

Compared to NDES, the DES strategy resulted in a mean gain of 1.42 life-years (95% CI: 1.14; 1.73) in terms of average life-years gained (LYG) (Table 2).

### 2.4. Probabilistic Cost-Effectiveness Analysis

The incremental cost per additional cure with DES versus NDES was estimated at EUR 212,493 (95% CI: EUR 15,493; EUR 634,462).

The incremental cost per life-year gained (LYG) with DES compared to NDES was EUR 24,618 ± EUR 19,535 (95% CI: EUR 1755–EUR 73,488) (Table 2). The probability that DES is cost-effective at a willingness-to-pay (WTP) threshold of <EUR 25,000 and <EUR 30,000 per LYG was 61.1% and 100%, respectively.

## 3. Discussion

This study presents a cost-effectiveness analysis of beta-lactam DES in the management of PAPs, a population often excluded from optimal antimicrobial therapy due to penicillin allergy labels [1,5,6]. The DES strategy was more effective than NDES in terms of both cure and survival. This finding suggests that DES may represent a clinically advantageous and economically sustainable strategy, potentially improving clinical outcomes while optimizing resource use within hospital settings.

The economic model developed here estimates a favorable cost-effectiveness profile for DES, with a 100% probability of being cost-effective at a WTP threshold of EUR 30,000 per LYG, and a 61.1% probability at the more conservative threshold of EUR 25,000. These figures are notable, as they suggest that DES not only aligns with established clinical goals (improving survival and cure rates) but may also fall within acceptable thresholds of health system affordability, particularly in publicly funded settings like the Spanish National Health System, where WTP thresholds are commonly estimated between EUR 21,000 and EUR 25,000 per quality-adjusted life year (QALY) gained [22].

To our knowledge, this is the first cost-effectiveness analysis conducted to evaluate DES in hospitalized PAPs. Among the main strengths of this analysis is the fact that cost data were derived from real-world clinical practice—a case–control study that collected effectiveness and cost data between 2015 and 2022 [15]. Another strength lies in the robustness of the economic model: in the probabilistic analysis, the probability of DES being cost-effective was 100% at a willingness-to-pay threshold of EUR 30,000 per LYG, although this probability dropped to 61.1% at the EUR 25,000 threshold.

Nevertheless, several limitations must be acknowledged when interpreting the results. First and foremost is the limited sample size (N = 56) of the source study, which inherently restricts the statistical power to detect significant differences in clinical outcomes. As stated in Rodríguez-Alarcón et al. study, sample size calculation was not considered feasible and a ratio 1:3 was selected for increasing statistical precision, as has been carried out in clinical trials involving rare diseases [15]. While a trend toward lower 30-day mortality and higher clinical cure rates was observed in the DES group compared to the NDES group, these differences did not reach statistical significance and were previously discussed in the source study [15]. This likely reflects a Type II error due to the small cohort, as previously noted in the methodological literature [23]. Despite this, as beta-lactam desensitization may be associated with increased costs due to the need to be performed in critical care units where patients are closely monitored during the time of procedure (usually 4–6 h), we considered relevant to conduct a cost-effectiveness model to clarify this issue and increase knowledge about this strategy, as the benefits of using first-line beta-lactam antibiotics in PAPs are considerable [1,3,5,6,7,8,9,10,11]. Furthermore, the lack of randomization and the retrospective design raise the possibility of residual confounding. Another limitation lies in the fact that long-term follow-up of the patients could not be carried out, a situation that could either overestimate or underestimate the results of the study. The economic model applied has been based on the experience applied in rare diseases and, in others such as oncological diseases, where cohorts that are usually compared do not reach statistically significant differences.

Although cost–utility analysis may be the method of choice in some circumstances, given that quality-adjusted life years (QALYs) encompass both the quantity and quality of life, in our study we opted solely for a cost-effectiveness analysis, which analyzed the cost of gaining a year of life with the most effective option. This was carried out for the following reasons: (i) The clinical study on which the economic analysis is based provides survival data, but not utility data, as patients’ quality of life data was not possible to obtain due to retrospective nature of the study. Consequently, the robust data for the economic analysis are those obtained directly from the Spanish patients in the clinical study; and (ii) Since the clinical study did not collect utility data and no data are available on Spanish patients in similar circumstances, introducing QALYs into the analysis would require making assumptions based on data from patients in other countries. This would weaken the internal validity of the economic model. Therefore, it was considered more appropriate to calculate the cost per life-year gained.

There was notable variability in prognostic factors (Charlson index, respiratory or renal disease) between the DES and NDES groups, which may explain the substantial cost difference: EUR 53,665 per patient in the DES group versus EUR 14,102 in the NDES group [15]. This cost gap might reflect underlying comorbidities rather than the DES procedure itself. This suggests that DES may have been reserved for patients with more severe infections or higher clinical risk, further complicating direct comparisons between groups. In fact, the multivariate analysis performed in Rodríguez-Alarcón et al.’s study reflects that hospital readmission rate and length of hospital stay were related to age and the Charlson index and were not independently associated with desensitization; therefore, the difference in resource consumption may be related to the baseline characteristics of patients [15]. It would be interesting to conduct further studies with larger sample sizes that include a multivariate sensitivity analysis to assess the effect of confounding.

Moreover, while the model assumes that improved access to beta-lactam antibiotics through DES leads to better outcomes and reduced long-term costs, this assumption requires confirmation in prospective, adequately powered studies. In addition, this study is based on single-center Spanish data which may limit the generalizability of the findings to other healthcare systems with different hospital cost structures, drug pricing, and antimicrobial stewardship policies. Large-scale multicenter trials or real-world implementation studies with standardized criteria for DES eligibility would be instrumental in validating these findings and refining the economic model. It would also be useful to evaluate additional outcomes, such as antimicrobial resistance patterns, readmission rates, and patient-reported quality of life.

In conclusion, despite the limitations inherent in modeling and retrospective data analysis, the present study provides preliminary evidence supporting the cost-effectiveness of beta-lactam DES in hospitalized high-risk PAPs with a confirmed allergy or in a compromised clinical situation when alternative treatment is not available or is considered suboptimal. Given the growing emphasis on value-based healthcare and antimicrobial stewardship, DES represents a promising strategy to improve the quality and efficiency of infectious disease management. The observed results should be verified in other future studies that could include a larger number of patients.

## 4. Materials and Methods

A cost-effectiveness analysis was conducted using a probabilistic decision-analytic model, incorporating 1000 s-order Monte Carlo simulations to account for parameter uncertainty [23,24]. Cost variables (continuous) were modeled using gamma distributions, while probability variables (dichotomous) were modeled using beta distributions, based on the minimum and maximum values obtained from the observational study [15], in accordance with established methodological guidelines [25,26]. A deterministic analysis of the per-patient cost difference was also performed.

The analysis adopted the perspective of the Spanish National Health System, including only direct hospital costs. Both cost data (in 2025 Euros) and effectiveness outcomes (cure and survival rates) for DES and NDES were derived from a retrospective cohort study by Rodríguez-Alarcón et al. [15].

DES-related costs included antibiotic acquisition, preparation materials (infusion bags, diluents, syringes), pharmacy personnel time (pharmacy technician), and a 4 h intensive care unit (ICU) stay, based on the estimated daily ICU cost of EUR 1250 in Spain [15,27].

In the deterministic analysis, the following outcomes were compared between the DES and NDES groups: (i) mean cost per patient; (ii) cure rate; and (iii) survival rate (Table 1). The probabilistic sensitivity analysis (PSA) results were reported as (i) the mean cost savings per patient (with 95% confidence intervals); (ii) probability of cost savings with DES; (iii) LYG per patient (mean and 95% CI); (iv) incremental cost per LYG (mean and 95% CI); and (v) probability of cost-effectiveness at two (WTP) thresholds (EUR 25,000 and EUR 30,000 per LYG) (Table 2). An additional outcome estimated in the PSA was the incremental cost per additional cure with DES compared to NDES. Table 1 includes the alpha and beta parameters used in probabilistic modeling.

Life expectancy gains were estimated using the mean patient age from the observational cohort (73.3 years) [15] and life expectancy for men and women in Spain (80.3 and 85.8 years, respectively), as reported by the National Institute of Statistics [22].

## 5. Conclusions

According to the present study, beta-lactam desensitization (DES) appears to be a cost-effective strategy for the treatment of selected PAPs. These results should be interpreted with caution due to the exploratory nature of the study, which employed data from a previously published study that did not demonstrate statistically significant differences, likely due to the limited sample size. Further validation in clinical studies with larger populations is warranted.

## Figures and Tables

**Table 1 antibiotics-14-00646-t001:** Cost and effectiveness data used in the economic model.

Variable	DES ^1^ (Mean)	NDES ^2^ (Mean)	Difference	Min Value	Max Value	SD	Distribution	Alpha	Beta
Cost per patient (EUR)	53,665	14,102	39,563	11,102	145,441	34,270	Gamma	1.3	29,685
Cure rate	0.714	0.548	0.166	0.133	0.199	0.017	Beta	79.9	401.6
Survival rate	0.857	0.714	0.143	0.114	0.172	0.015	Beta	82.2	492.4

^1^ DES: beta-lactam desensitization; ^2^ NDES: no desensitization.

**Table 2 antibiotics-14-00646-t002:** Cost-effectiveness results for beta-lactam desensitization in PAPs.

Item	Result
Additional cost per patient (DES vs. NDES) ^1^	EUR 37,805 (95% CI: EUR 2023–EUR 126,785)
Life-years gained (LYG) per patient (DES vs. NDES) ^2^	1.42 (95% CI: 1.15–1.73)
Cost per LYG (DES vs. NDES) ^3^	EUR 24,618 ± EUR 19,535 (95% CI: EUR 1755–EUR 73,488)
Probability of cost-effectiveness (cost/LYG < EUR 25,000)	61.1%
Probability of cost-effectiveness (cost/LYG < EUR 30,000)	100%
^1^ DES vs. NDES: beta-lactam desensitization vs. no desensitization	

^1^ Additional cost per patient: the mean increase in direct hospital costs incurred when treating a PAP using beta-lactam DES, compared to standard NDES. ^2^ Life-years gained per patient: the average number of additional years of life expected per patient as a result of receiving DES compared to NDES. ^3^ Cost per life-year gained: the incremental cost-effectiveness ratio calculated as the additional cost required to gain one extra year of life by using DES instead of NDES.

## Data Availability

The original contributions presented in this study are included in the article. Further inquiries can be directed to the corresponding author.

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
