# Peer review of "The Cost-Effectiveness of Beta-Lactam Desensitization in the Management of Penicillin-Allergic Patients"

_antibiotics, 2025, doi:10.3390/antibiotics14070646_

Round 1

Reviewer 1 Report

Comments and Suggestions for Authors

This manuscript examines the cost-effectiveness of beta-lactam desensitisation (DES) in penicillin-allergic hospitalised patients, utilising a retrospective case-control dataset comprising 56 patients and a probabilistic economic model. While the topic is clinically relevant, the study’s results exhibit significant uncertainty. The conclusions exceed the strength of the evidence.
The manuscript is generally well-written. The introduction provides an adequate overview of the problem and current strategies, including DES. The results are clearly organized, with appropriate tables. Nonetheless, some improvements are recommended in the Discussion and Conclusions sections:

  • Add a paragraph in the Discussion section to justify the absence of the QALY (Quality-Adjusted Life Years) index, the standard outcome measure in health economic evaluations.
  • Adjust the conclusions to reflect the exploratory nature of the study. Since the p-value for the difference in survival is far from the conventional threshold for significance, it is incorrect to state these differences are clinically meaningful without stronger evidence. Adjust the conclusions to reflect the exploratory nature of the study.

Reviewer 2 Report

Comments and Suggestions for Authors

Dear author,

The manuscript addresses an important issue in antimicrobial stewardship by evaluating the cost-effectiveness of beta-lactam desensitization in penicillin-allergic patients. While the topic is clinically relevant, several concerns regarding the study design, methodology, and interpretation of the results must be addressed.

  1. The article needs to be organized according to the Journal guidelines.
  2. The study includes only 14 patients in the DES group, which limits the statistical power and raises the possibility of Type II error. Have the authors performed any power analysis to determine whether the sample size is adequate to detect clinically meaningful differences?
  3. The cure rate and 30-day mortality differences between groups did not reach statistical significance. Given that these outcomes form the foundation of your economic model, how do you justify modeling cost-effectiveness based on non-significant clinical effects?
  4. DES appears to have been applied to higher-risk patients (with higher Charlson index or comorbidities), possibly inflating the costs and biasing the comparison. Please clarify whether any adjustment for baseline differences was performed, and if not, consider including a multivariate sensitivity analysis to assess the effect of confounding.
  5. The model uses life-years gained (LYG) but does not incorporate quality-adjusted life years (QALYs), which are typically used in health economic evaluations. Why did you choose not to use QALYs in your model? Would incorporating utility weights significantly alter the cost-effectiveness thresholds?
  6. Direct hospital costs are used, but indirect costs (e.g., readmissions, long-term outcomes, adverse event management) are not included. Could the authors explain the rationale for excluding broader health economic perspectives? Do you believe this underestimates or overestimates the true cost-effectiveness of DES?
  7. The data is derived from a single-center Spanish cohort. How generalizable do the authors believe these findings are to other healthcare systems with different cost structures and antimicrobial policies?
  8. The conclusion suggests DES is a cost-effective strategy; however, the low probability (61.1%) of cost-effectiveness at €25,000 per LYG introduces uncertainty. Would the authors consider rephrasing the conclusions to reflect the need for cautious interpretation and further validation?
  9. Revise abbreviations briefly and mention them in their first place.
  10. Typo in Abstract line 26: “additional cost with DES was €37,805” consider clarifying this as “incremental cost”
  11. Please improve the language and sentence structure in the Discussion and Conclusion sections to make them more solid and impactful. The conclusion should convey the key findings and implications of the study to the reader.
  12. A thorough English proofreading is recommended to enhance clarity and readability throughout the manuscript.

Suggestion: If possible, the authors should consider visualizing the data. Including a flow diagram of the study design or graphical representations of the results would help readers better understand the methodology and outcomes.

Comments on the Quality of English Language

English proofreading is recommended to enhance clarity and readability throughout the manuscript.

Round 2

Reviewer 1 Report

Comments and Suggestions for Authors

I appreciate the careful revisions made by the authors. All concerns have been satisfactorily solved. The revised manuscript is suitable for publication.

Reviewer 2 Report

Comments and Suggestions for Authors

Dear Authors,

I believe you have adequately addressed all the questions and provided sufficient responses. The manuscript has been notably improved. I recommend it for acceptance and further processing.